# Effects of *Lactobacillus salivarius* LN12 in Combination with Amoxicillin and Clarithromycin on *Helicobacter pylori* Biofilm In Vitro

**DOI:** 10.3390/microorganisms9081611

**Published:** 2021-07-28

**Authors:** Fang Jin, Hong Yang

**Affiliations:** State Key Laboratory of Microbial Metabolism, School of Life Sciences and Biotechnology, Shanghai Jiao Tong University, Shanghai 201100, China; jinfang@sjtu.edu.cn

**Keywords:** *Lactobacillus salivarius*, *Helicobacter pylori*, biofilm, clarithromycin, amoxicillin

## Abstract

*Helicobacter pylori* is a highly prevalent and harmful gastrointestinal pathogen. Antibiotic resistance and biofilm complexity have led to a decrease in the cure rate. Probiotics are considered to be an adjuvant therapy for clinical *Helicobacter pylori* infections. However, there is no substantial explanation for the adjuvant role of probiotics on *H. pylori* biofilm. In this study, the effects of probiotics in combination with amoxicillin (AMX) and clarithromycin (CLR) on *H. pylori* biofilms were explored in vitro for the first time. The minimum inhibitory concentration (MIC) and the fractional inhibitory concentration (FIC) for *H. pylori* was determined by the microbroth dilution method, and the plate counting method was used to determine the minimum biofilm removal concentration (MBEC) and survival rate for *H. pylori* biofilm. The biofilm structure was observed by scanning electron microscopy (SEM) and confocal laser scanning microscopy (CLSM), protein and polysaccharide contents in extracellular polymeric substances (EPS) were determined by the Bradford method and the phenol-sulfate method, respectively. The gene expression levels of *cagA* and *vacA* were evaluated by real-time qPCR. Among the ten *H. pylori* strains, the clinical strain 3192 showed the strongest film-forming ability, the 3192 biofilms significantly improved the resistance to AMX and CLR, and AMX and CLR showed antagonistic effects on planktonic 3192 cells. When the *Lactobacillus salivarius* LN12 cell-free supernatant (CFS) was in combination with AMX and CLR, the 3192 biofilm structure was destroyed to a greater extent than when separately; more biofilm biomass and protein in EPS was decreased; and the downregulation effect of the virulence gene *vacA* was also greater than that of single use. In this study, we suggest that the addition of LN12 to AMX and CLR may enhance the therapeutic effect of triple therapy, especially for the treatment of *H. pylori* biofilms.

## 1. Introduction

*Helicobacter pylori* (*H. pylori*) is a spiral-shaped Gram-negative pathogen that colonizes the stomach in approximately half of the world’s population. It can spread from person to person and is closely related to peptic ulcer disease, atrophic gastritis, gastric adenocarcinoma, and MALT. Although *H. pylori* prevalence varies from country to country, the infection rate in developing countries is relatively high (80–90%). It was designated as a class I carcinogen by the World Health Organization (WHO) in 1994 [1,2]. The Maastricht V/Florence Consensus Report and Chinese Fifth National Consensus Report on the Treatment of *H. pylori* Infection both recommend that all patients infected with *H. pylori* undergo radical treatment [3]. The treatment usually consists of a proton pump inhibitor and two to three antibiotics (amoxicillin (AMX), clarithromycin (CLR), metronidazole, levofloxacin, and tetracycline), and the treatment cycle is 10–14 days. A meta-analysis in 2018 showed that clarithromycin, metronidazole, and levofloxacin resistance rates exceeded 15% (alarm level) in all regions of the World Health Organization [4]. Standard triple therapy (proton pump inhibitor + clarithromycin + amoxicillin) was first proposed at the Maastricht conference and has been widely used in clinical treatment for more than 20 years. However, with the emergence of clarithromycin resistance, the current eradication rate of this therapy is less than 80% [5]. In 2017, *H. pylori* resistance to clarithromycin was defined as a high priority bacterium in the WHO’s priority list of resistant bacteria [6]. In addition, the continuous discovery of multidrug resistant strains makes antibiotic resistance a major challenge in the treatment of *H. pylori*.

The spiral shape and flagella sheath on flagellum make colonization under low pH conditions more powerful for *H. pylori. H. pylori* can produce urease to neutralize the acidic environment in the stomach for survival; superoxide dismutase and catalase to protect against neutrophil killing; and adhesion proteins (BabA, SabA, HopZ, AlpA/B, etc.) to settle in the epithelial cells [7]. Vacuolating cytotoxin protein (VacA) and cytotoxin-associated protein (CagA) are the main virulence factors [8]. CagA plays an important role in the injection of bacterial components into the host epithelium, it can trigger the production of IL-8 and INF-γ, and ultimately destroy the epithelial barrier [9]. VacA can change the antigen presentation of B cells and inhibits the proliferation of T cells, making it an important virulence factor for the establishment of chronic infections, and VacA can damage the gastric mucosa and delay the repair of gastric epithelium [10].

Biofilms play a vital role in resisting external threats, and *H. pylori* has the ability to form biofilms in vivo and in vitro [11,12]. The matrix of *H. pylori* biofilms contains mannan and LPS-related structures, extracellular DNA (eDNA), protein, and outer membrane vesicles (OMV) [13]. Studies have shown that *H. pylori* biofilms increase clarithromycin, amoxicillin, and metronidazole resistance in vitro, and the expression of proton pump genes related to antibiotics is increased 10–1000 times compared with the planktonic state [11,14]. Studies have compared the relationship between biofilm formation ability and antibiotic resistance in 106 clinical *H. pylori* strains and found that there was no significant difference in biofilm formation ability between antibiotic-resistant and antibiotic-sensitive strains [15].

Antibiotic resistance and biofilms both limit the effectiveness of antibiotics against *H. pylori* infection. In recent years, some natural products (natural capsaicin, nutmeg, olive, etc.) and probiotics (*Lactobacillus reuteri*, *Lactobacillus casei*, *Bifidobacterium longum,* etc.) have shown therapeutic effects on *H. pylori* in vitro and in vivo [16,17,18,19]. As an important part of the stomach and intestinal flora, probiotics can not only be used for treating *H. pylori*, but also reduce the disturbance of the gastrointestinal flora caused by antibiotics and alleviate the side effects of antibiotics. Probiotics can affect *H. pylori* growth, reduce the pH value in the cavity, compete for adhesion sites and nutrients, and produce bacteriocins to destroy *H. pylori* biofilms [20]. Probiotics will not produce strong selective pressure on drug-resistant strains as traditional antibiotics do, and their cytotoxicity is also lower than that of biofilm quorum-sensing inhibitors, so they can be considered as ideal choices for new anti-biofilms [20]. At present, probiotics are often used as an adjuvant agent to antibiotics for *H. pylori*, but some doctors are skeptical about the effect of probiotics. This is mainly due to the lack of theoretical foundation and understanding of the mechanism of action of this combination therapy. The strain, time, and duration of probiotics in treatment are still unclear [21].

In this study, clinically isolated *H. pylori* strain 3192 was used as the research object. Clarithromycin and amoxicillin showed antagonistic effects on planktonic 3192 in vitro, and 3192 had a strong biofilm-forming ability. The effects of the combined use of the clarithromycin, amoxicillin, and *Lactobacillus salivarius* LN12 cell-free supernatant (CFS) on 3192 biofilms were evaluated in vitro for the first time in order to provide a basis for clinical treatment of *H. pylori.*

## 2. Materials and Methods

### 2.1. Bacterial Strain, Medium and Growth Conditions

*H. pylori* SS1 and eight other clinical strains were donated by the Shanghai Renji Hospital (Shanghai, China), and *H. pylori* ATCC43504 was purchased from American Type Culture Collection (ATCC) (Manassas, VA, USA). For solid culture, *H. pylori* was inoculated on Columbia blood agar plates containing 5% sterile defibrinated sheep blood for 48–72 h at 37 °C under microaerobic conditions (MGC, Tokyo, Japan). For liquid culture, *H. pylori* was cultured in Brucella broth (BB) containing different concentrations of Gibco fetal serum (FBS) (10% for growth; 2% for biofilm) for 2–4 d at 37 °C and 80–100 rpm in microaerobic conditions. The cryopreserved strains were stored in BB containing 10% FBS and 20% glycerol at −80 °C.

All twenty-eight strains of probiotics were provided by Jiaxing Innocul-Probiotics Co. Ltd. (Jiaxing, Zhejiang, China), and modified de Man Rogosa Sharpe broth (mMRS) (tryptone 10 g, beef extract powder 10 g, yeast extract 5 g, glucose 20 g, K_2_HPO_4_ 2 g, ammonium citrate 2 g, MgSO_4_·7H_2_O 0.2 g, MnSO_4_·4H_2_O 0.2 g, and CH_3_COONa 1.5 g brought to a volume of 1 L with distilled water) was used for culturing probiotics for 12–24 h at 37 °C aerobically. Probiotics included the following species: *Bifidobacterium bifidum* (*n* = 2), *Bifidobacterium animalis* (*n* = 3), *Bifidobacterium longum* (*n* = 7), *Bifidobacterium infantis* (*n* = 1), *Bifidobacterium breve* (*n* = 1), *Lactobacillus helveticus* (*n* = 2), *Lactobacillus plantarum* (*n* = 1), *Lactobacillus gasseri* (*n* = 1), *Lactobacillus fermentum* (*n* = 1), *Lactobacillus rhamnosus* (*n* = 1), *Lactobacillus salivarius* (*n* = 1), *Lactobacillus acidophilus* (*n* = 1), *Lactobacillus casei* (*n* = 1), *Lactobacillus paracasei* (*n* = 2), and *Lactobacillus reuteri* (*n* = 3).

### 2.2. MIC and FIC of Antibiotics

The minimum inhibitory concentration (MIC) of *H. pylori* for amoxicillin, clarithromycin, levofloxacin, tetracycline, and metronidazole was slightly modified by the microbroth dilution method, and ATCC43504 was used as a control [22]. Antibiotics were diluted with BB10 (BB containing 10% FBS) to different concentrations, and *H. pylori* was prepared with BB10 to 10^8^ CFU/mL and inoculated with antibiotics in 96-well polyethylene cell culture plates with a total volume of 200 μL per well. Plates were cultured at 37 °C and 100 rpm under microaerobic conditions for at least 72 h, and the OD_600nm_ was read with a microplate reader. The concentration of the bacterial solution in each well was no less than 10^6^ CFU/mL. BB10 and the bacterial solution were used as negative and positive controls, respectively, and the MIC was defined as the lowest drug concentration that could inhibit the growth of *H. pylori*. According to the recommendations of the European Antimicrobial Susceptibility Testing Committee (EUCAST), the cutoff values of MIC for amoxicillin, clarithromycin, levofloxacin, tetracycline, and metronidazole are >0.125, >0.5, >8, >1, and >1 mg/L, respectively [23].

A checkerboard assay was used to determine the synergistic effect of amoxicillin and clarithromycin on *H. pylori*. The two antibiotics were diluted horizontally and vertically in 96-well microtiter plates. *H. pylori* was diluted with BB10 to 10^8^ CFU/mL and inoculated with different concentrations of antibiotics in 96-well plates according to a total volume of 200 μL per well. The control group and the cultivation and measurement parameters were the same as above. The fractional inhibitory concentration (FIC) of amoxicillin and clarithromycin was calculated according to the following formula:(1)$$\mathrm{FIC}\;=\frac{\mathrm{MIC}\left(A\;\mathrm{in}\;\mathrm{combnation}\;\mathrm{with}\;B\right)}{\mathrm{MIC}\left(A\;\mathrm{alone}\right)}+\frac{\mathrm{MIC}\left(B\;\mathrm{in}\;\mathrm{combnation}\;\mathrm{with}\;A\right)}{\mathrm{MIC}\left(B\;\mathrm{alone}\right)}$$

A FIC value of 0.5 or below and 0.75, 1.0, and 2.0 or above is defined as partial synergy, additive, independent, and antagonistic, respectively [24].

### 2.3. Construction of H. pylori Biofilm

The *H. pylori* biofilm formation method was performed as described by Yu et al. with slight modifications [25,26]. *H. pylori* was resuspended in BB2 (BB containing 2% FBS) to 10^7^ CFU/mL, and 2 mL of bacterial suspension was inoculated into the wells of a 12-well sterile cell culture plate with a 20 mm × 20 mm borosilicate coverslip (Matsunami Glass, Tokyo, Japan) in each well. The plate was placed at 37 °C at 100 rpm for four days under microaerobic conditions, and the biofilm appeared at the gas–liquid interface.

### 2.4. H. pylori Biofilm Resistance to Antibiotics

The method for determining the sensitivity of *H. pylori* biofilm cells to antibiotics was slightly modified from the method of Hideo et al. [11,15]. The 4-d biofilm was washed three times with PBS and then placed in various concentrations of amoxicillin and clarithromycin in a 12-well plate, with BB2 serving as a negative control. After 24 h under microaerobic conditions, the biofilm was scraped off the glass cover and resuspended in 1 mL of PBS, and the colony forming units (CFUs) were measured on Columbia blood agar plates.

### 2.5. Zone of Inhibition on H. pylori

The antibacterial effect of probiotics on *H. pylori* was determined by the Oxford Cup experiment [27]. *H. pylori* grown on Columbia blood agar plates was collected and washed with PBS twice, and the 10^8^ CFU/mL solution was spread on Columbia blood agar plates. After growing to 10^9^ CFU/mL in mMRS, LN12 was centrifuged at 7600 rpm at 4 °C for 10 min, CFS was obtained by filtration through a 0.22 μm filter membrane, 100 μL of the supernatant was injected into an Oxford cup, and mMRS was used as the negative control. After 72 h of microaerobic culture at 37 °C, the inhibition diameter was measured.

### 2.6. MIC of LN12 CFS

The method for determining the MIC of the *Lactobacillus salivarius* LN12 supernatant (CFS) against *H. pylori* refers to that published by Piotrowski [28]. LN12 CFS was obtained as above and diluted with BB10. *H. pylori* was resuspended in BB10 and inoculated into a 96-well polyethylene cell culture plate containing 5–50% serially diluted CFS in a total volume of 200 μL per well. The cell concentration per well was not less than 10^6^ CFU/mL, and after incubating at 37 °C and 100 rpm microaerobic conditions for at least 72 h, the OD_600nm_ was measured with a microplate reader to determine the MIC of LN12 CFS.

### 2.7. Different Treatments on the Effectiveness of LN12 CFS

The effects of LN12 CFS on *H. pylori* biofilms under different treatments were explored according to Ji [29]. LN12 CFS was divided into four groups, namely, (1) treated with 1 mg/mL proteinase K (final concentration) for 1 h at 37 °C and inactivated by heating at 95 °C for 1 min (proteinase K-treated CFS); (2) treated with 0.5 mg/mL catalase (final concentration) for 1 h at 37 °C and inactivated by heating at 95 °C for 30 min (catalase-treated CFS); (3) adjusted the pH to 6.5 with 1 M NaOH (neutralized CFS); and (4) 115 °C heated for 20 min (heat-treated CFS), BB2 was the control, and after 24 h of the different treatments, the biofilms were resuspended in 1 mL of PBS with a cell scraper and the viability was compared by counting CFUs on Columbia blood agar plates [30,31].

### 2.8. The Effect of LN12 CFS Combined with Antibiotics on the Biomass of H. pylori Biofilm

After washing, the 4-d mature *H. pylori* 3192 biofilm was treated differently in a 12-well polyethylene cell culture plate. For the amoxicillin and clarithromycin treatment group (FIC group) or the LN12 CFS treatment group (CFS group), 2 mL of 1 × FIC concentration of amoxicillin and clarithromycin or 1/2 × MIC of LN12 CFS diluted with BB2 was added to the 12-well plate. The combination of amoxicillin, clarithromycin, and LN12 CFS (CF group) contained both 1 × FIC of amoxicillin and clarithromycin and 1/2 × MIC of LN12 CFS. After 24 h under microaerobic conditions, the biofilm biomass was quantified by crystal violet. After removing the culture medium, the cover glass was washed twice with PBS, stained with 1% crystal violet solution for 15 min, washed twice with PBS, redissolved in 400 μL of 95% ethanol, and the absorbance was measured at 595 nm.

### 2.9. The Effect of LN12 CFS Combined with Antibiotics on the Morphology and Structure of H. pylori Biofilm

Scanning electron microscopy (SEM) was used to observe the morphology and structure of *H. pylori* biofilms according to Fauzia et al. [15]. The 3192 biofilms under different treatments were washed with PBS and fixed in 50% glutaraldehyde overnight. After dehydration with a gradient of 50–100% ethanol, biofilms were dried at the supercritical point. After spraying with gold (Leica, Wetzlar, Germany), SEM analysis was performed on an scanning electron microscope (S-4800, Hitachi, Tokyo, Japan).

### 2.10. The Effect of LN12 CFS Combined with Antibiotics on the Survival Rate of H. pylori Biofilm

Confocal laser scanning microscopy (CLSM) was used to evaluate the changes in the survival rate of the biofilm under different treatments according to Yu and Li [22]. After different treatments as above, the biofilm was scraped off with a cell scraper and resuspended in 1 mL of PBS, and CFUs were compared on plates. The biofilms of different treatments were carefully washed twice with 0.9% NaCl solution, placed in a 1:1 mixture of SYTO 9 (staining live cells) and propidium iodide (staining dead cells) (Invitrogen, Carlsbad, CA, USA) for 20 min at room temperature away from light, and washed with 0.9% NaCl twice. The biofilm was observed under an upright laser confocal microscope (Nikon, Tokyo, Japan) with an oil microscope. The excitation wavelengths were 488 and 560 nm, and the Z-axis acquisition signal step length was 0.1 μm [25,32]. The collected images were measured and analyzed using Nikon Ni-E A1 HD25 special software.

### 2.11. Effect of LN12 CFS Combined with Antibiotics on EPS of H. pylori Biofilm

CLSM observations and quantitative analysis were combined to compare the effects of different treatments on the extracellular polymeric substances (EPS) of biofilms. The biofilms after different treatments were stained with FITC-ConA (500 μg/mL) (Sigma-Aldrich, St. Louis, MO, USA) under dark conditions for 30 min and cleaned with 0.9% NaCl solution. CLSM was used to observe the 488-channel wavelength setting. The biofilms of different treatments were scraped with a cell scraper, resuspended in 1 mL of PBS, and then centrifuged at 4 °C at 10,000 rpm for 30 min. The supernatant was collected for the determination of the content of polysaccharides and protein substances in EPS, the content of polysaccharides was determined by the sulfuric acid phenol method, and the content of protein was determined by the Bradford Protein Concentration Determination Kit (Sangon, Shanghai, China).

### 2.12. The Effect of LN12 CFS Combined with Antibiotics on the Expression of Virulence Genes of H. pylori Biofilm

The total RNA of the biofilm under different treatments was extracted using the TransZol Up Plus RNA Kit (TransGen Biotech, Beijing, China) and cDNA was synthesized by the PrimeScript™ RT Reagent Kit with gDNA Eraser (Perfect Real Time) (Takara, Kyoto, Japan). Finally, qPCR was used to assess relative gene expression with TB Green Premix Ex TaqTM II (Takara, Kyoto, Japan) on a quantitative PCR apparatus (Analytikjena, Jena, Germany). The primers used in the experiment are listed in Table 1. The expression level of each gene was normalized with the expression level of 16S rRNA as an internal reference, and the relative expression level was calculated using the 2^−^^∆∆CT^ method.

### 2.13. Statistical Analysis

Statistical analyses and graphs were generated by Originpro Learning Edition (OriginLab, Northampton, MA, USA). For all experiments, the data were presented as the mean ± standard deviation (*n* = 3) and were analyzed using one-way analysis of variance (ANOVA) followed by Tukey’s multiple comparison test. A *p* < 0.05 was considered as a statistically significant difference.

## 3. Results

### 3.1. MIC of Antibiotics and Lactobacillus salivarius LN12 Cell-Free Supernatant (CFS)

The MICs of amoxicillin, clarithromycin, levofloxacin, tetracycline, and metronidazole for ten *H. pylori* strains are shown in Table 2. SS1 and ATCC43504 were used as control strains, and the other eight strains were clinically isolated. Twelve percent of all strains were resistant to amoxicillin, 50% were resistant to clarithromycin, 50% were resistant to levofloxacin, 87.5% were resistant to metronidazole, and 0% were resistant to tetracycline. Multidrug resistance means that one strain is resistant to two or more antibiotics. Of the eight clinical strains, 25% were resistant to two antibiotics, and 50% were resistant to three antibiotics.

When *Lactobacillus salivarius* LN12 was cultured to a cell concentration of approximately 1 × 10^9^ CFU/mL, CFS was collected. The lowest dilution with BB10 that could inhibit the growth of *H. pylori* was 12.5%, so the MIC of LN12 CFS was defined as 12.5%.

### 3.2. FIC of Amoxicillin and Clarithromycin for H. pylori

The FIC is defined by the checkerboard microdilution method as partial synergistic (0.5 < FIC ≤ 0.75), additive (0.75 < FIC ≤ 1.0), independent (1.0 < FIC ≤ 2.0), and antagonistic (FIC > 2.0) [24]. The FICs of amoxicillin and clarithromycin against ten strains of *H. pylori* were determined as shown in Table 3. The combination of amoxicillin and clarithromycin had different combined effects on different *H. pylori* strains. Among all strains, partial synergy occurred in 30%, addition in 20%, independence in 30%, and antagonistic in 20% (clinical strains 3192 and 3931).

### 3.3. H. pylori Biofilm

When comparing the biofilm-forming abilities of the ten *H. pylori* strains by crystal violet staining, an OD_595nm_ of 0.3 was regarded as the dividing line with reference to Fauzia [15]. Strains with OD_595nm_ greater than 0.3 were regarded with a stronger ability to form biofilms, where lower than OD_595nm_ 0.3 had weaker biofilm formation ability. *H. pylori* 3192 and 3750 had significantly stronger biofilm formation ability than the other strains. The 3192 strain had the strongest biofilm formation ability, as shown in Figure 1. The biofilms formed by 3192, 3750, 4386, and SS1 are shown in Figure 2.

### 3.4. MBEC of Amoxicillin and Clarithromycin for H. pylori 3192

Compared with the MIC, the minimum biofilm removal concentration (MBEC) can more accurately reflect the resistance of biofilm structure to antibiotics. To determine the MBEC of the 3192 biofilm, the 4-d 3192 biofilm on the cover glass was treated with different concentrations of clarithromycin and amoxicillin, and the viable CFUs in biofilm were counted to observe the effects of different treatments. The initial CFUs of the 3192 biofilm was approximately 10^7^ CFU/mL, and the CFUs decreased significantly after treatment with 1 × MIC, 4 × MIC, 16 × MIC, 64 × MIC, 128 × MIC, and 256 × MIC amoxicillin for 24 h, no CFUs were observed after 256 × MIC amoxicillin treatment for 24 h. To determine the MBEC of 3192 biofilms against clarithromycin, CFUs of viable bacteria were counted after treatment with 1 × MIC, 2 × MIC, 4 × MIC, and 8 × MIC CLR, no CFUs were found after 8 × MIC treatment for 24 h. The effects of clarithromycin (A) and amoxicillin (B) on cell viability of *H. pylori* 3192 biofilms are shown in Figure 3. The MBEC of 3192 to amoxicillin was 3.906 μg/mL (256 × MIC_AMX_), and the MBEC of clarithromycin was 0.061 μg/mL (8 × MIC_CLR_).

### 3.5. Zone of Inhibition on H. pylori

Twenty-eight probiotics were tested on two *H. pylori* reference strains, namely, SS1 and ATCC43504, and eight clinical isolates through the Oxford cup method. Seven probiotics had antibacterial effects, as shown in Table 4, and the results of strains without antibacterial effects were not shown. LN66 (*Lactobacillus plantarum*), LN12 (*Lactobacillus salivarius*), and LN19 (*Lactobacillus helveticus*) had significant antibacterial effects on ten strains of *H. pylori*. The antibacterial effects of LN12 and LN19 were weaker than those of LN66.

### 3.6. The Impact of Different Treatments of CFS on the Survival Rate of Biofilm

After evaluation, LN12 was superior to LN66 and LN19 in tolerance to simulated GIT conditions (Appendix A), auto-aggregation (supplementary Appendix A), and cell surface hydrophobicity (Appendix A) and was used for further exploration of 3192 biofilms. The probiotic fermentation broth was rich in metabolites. To clarify the substances that played a role in destroying the *H. pylori* biofilm, LN12 CFS was administered to four groups that were subjected to proteinase K, catalase, heat treatment, or neutralization, the untreated CFS group served as a positive control, and BB2 as negative controls. All groups were incubated with the 4-d biofilm for 24 h. The experimental results in Figure 4 show that the untreated LN12 CFS group had significant killing effects on *H. pylori* CFUs in biofilms, which was not caused by the LN12 medium (mMRS) but by metabolites. The four grouping treatments had different effects on the killing effects of LN12 CFS. The effect of heat treatment was not significantly different from that of CFS, indicating that the metabolites that played a role were thermally stable, and the neutralization treatment significantly weakened the killing effect of LN12. Some acids in LN12 CFS had a killing effect; after proteinase K treatment, the killing effect was significantly weakened, indicating that there were some antibacterial proteins in CFS. There was no significant difference between the results of catalase treatment and CFS, and it was believed that CFS did not contain hydrogen peroxide or contained too low a concentration of hydrogen peroxide.

### 3.7. The Effect of LN12 CFS with Antibiotics on the Morphology of H. pylori Biofilm

To compare the effects of LN12 CFS combined with amoxicillin and clarithromycin on the mature biofilm of 3192, the amoxicillin and clarithromycin concentrations were set to the 1 × FIC, the LN12 CFS concentration was set to 1/2 × MIC, and 1/2 × MIC of LN12 CFS and 1 × FIC of amoxicillin and clarithromycin were combined when used together. *H. pylori* 3192 formed biofilms at the air–liquid interface on the cover glass, and the effect of different combinations on the morphology of 3192 biofilm cells was observed by scanning electron microscopy, as shown in Figure 5. Under 5000× magnification, *H. pylori* in the control group was mainly arranged in a spiral shape. Under 10,000× magnification, a small number of 3192 cells could be observed as coccoid and flagella could be seen. It was observed that the biofilm contained bacteria and other substances. This is consistent with the literature reporting that the extracellular polymer is composed of extracellular DNA, extracellular protein, and OMV, which can act as a barrier to protect the bacterial community from immune cells and antibacterial agents in *H. pylori* [12]. After different treatments, it was found that the biofilm appeared looser and broken, and large pores appeared. When the probiotic supernatant was combined with amoxicillin and clarithromycin, the damage to the biofilm was the strongest, and the spiral shape was significantly reduced.

### 3.8. The Effect of LN12 CFS with Antibiotics on H. pylori Biofilm Biomass

The crystal violet staining method was used to compare the effects of LN12 CFS and amoxicillin and clarithromycin on the biomass of the 3192 mature biofilms when used alone or in combination. The concentration was prepared as above. Figure 6 shows that different treatments had significant effects on the biomass of the 3192 biofilms compared with the control group (*p* < 0.05). Amoxicillin, clarithromycin, and LN12 CFS alone can significantly reduce the biomass without a significant difference, but the effect was not as strong as the combined one, and the FIC group had better effects than the CFS group.

### 3.9. The Effect of LN12 CFS with Antibiotics on the Survival Rate of H. pylori Biofilm

The concentration and combination of CFS and antibiotics were determined in accordance with the above method. CFU counting on Columbia blood agar plates and CLSM were used to observe the survival rate and structure of biofilms. The CFU results showed that clarithromycin and amoxicillin, CFS alone, or their combination can kill biofilm cells, and the effect of CFS was significantly greater than that of the antibiotics, but the effects of antibiotics and CFS alone were not as good as those of the combination of the two. CFS in combination with amoxicillin and clarithromycin reduced the CFUs of the control group from 10^7^ CFU/mL to 10^4^ CFU/mL, as shown in Figure 7.

In the crystal violet test, the total biomass of the biofilm including live and dead bacteria and EPS was measured. The structure of the biofilm and the vitality of the internal bacteria can be visualized with CLSM. The biofilms after 24 h of different treatments were analyzed by confocal laser microscopy (Figure 8). The Live/Dead Kit can distinguish between live cells (green) and dead cells (red). Dense green fluorescent clusters were observed in the control group, indicating that the untreated biofilm had a compact and dense structure, the proportion of dead cells under CLSM was almost zero, and the thickness of the biofilm was 4.5 μm. The structure of the biofilm underwent different destruction after different treatments, and the average biofilm thickness was 3.9 μm for FIC, 3.5 μm for CFS, and 3.4 μm for CF. The x-y axis and the fixed angle view showed that the biofilm was dispersed to varying degrees. When the FIC and CFS were used alone, the number of viable bacteria was reduced. The degree of dispersion with the combined use of CFS was significantly greater than that of single use, and the structure was looser, which corresponded to the SEM and biofilm CFU results. The mature *H. pylori* biofilm was difficult to remove by a single antibiotic, and the biofilm kept the internal bacteria alive under the protection of the outer EPS. As shown in Figure 8, the red fluorescent part can be seen under different treatments, and the proportion of red fluorescence was as follows: FIC < CFS < CF.

### 3.10. Effect of LN12 CFS Combined with Antibiotics on EPS of H. pylori Biofilm

The *H. pylori* biofilm is composed of bacterial cells and the EPS matrix wrapped around the outside. EPS forms a three-dimensional structure of the biofilm scaffold and is responsible for adhesion, cell–cell interaction, and signal transmission in the process of biofilm formation. Since polysaccharides are the main part of the EPS matrix, FITC-ConA can specifically bind to the D-(+)-glucose and D-(+)-mannose of polysaccharides to observe the EPS matrix of biofilms. The EPS in the biofilm was stained with FITC-ConA and observed under CLSM in the 488 channel with green fluorescence. The effect of different treatments on the EPS of the *H. pylori* biofilm was observed through CLSM as shown in Figure 9. It can be seen from Figure 9 that *H. pylori* formed biofilms that were characterized by a rich EPS matrix aggregation. The polysaccharide structure in the biofilm was dense, and different treatments caused varying degrees of damage to the EPS structure. The destructive effect of CFS alone was weaker than that of antibiotics alone or the combination of the two.

The content of protein and polysaccharide in the *H. pylori* 3192 biofilm was calculated by the Bradford and sulfuric phenol methods, as shown in Figure 10. The results showed that for protein, the FIC did not significantly affect the protein contents, but CFS had a significant effect on it, and when the CFS and FIC were used together, the weakening effect was greater than CFS alone. For polysaccharides, the three treatments all significantly weakened the biofilm, and the effect of FIC was significantly stronger than that of CFS, but there was no significant difference between the combination of the two and FIC.

### 3.11. The Effect of LN12 CFS Combined with Antibiotics on the Expression of Virulence Genes of H. pylori Biofilm

The relative expression levels of the virulence genes *vacA* and *cagA* were determined after different treatments for 24 h, as shown in Figure 11. For the *cagA* gene, 1/2 MIC of LN12 CFS caused the transcription of the *cagA* gene to be upregulated, and 1 × FIC of amoxicillin and clarithromycin caused the transcription of the *cagA* gene to be downregulated. When the two were used in combination, there was no significant difference compared to the control group; for the *vacA* gene, 1/2 × MIC of CFS and 1 × FIC of amoxicillin and clarithromycin both caused the transcription to be downregulated, and the downregulation of CFS was more significant. The downregulation effect of the combined use was greater than that of the single use. Many interactions have occurred during the formation of microbial biofilms, and it is difficult to clearly assess the usefulness or unavailability of *H. pylori* genes during the formation of this complex structure [13].

## 4. Discussion

*H. pylori* is the main pathogen that causes chronic gastritis, peptic ulcers, atrophic gastritis, and other gastric diseases and is classified as a category I carcinogen. To eradicate *H. pylori* infection, the most widely recommended first-line treatment strategy is the standard triple therapy consisting of amoxicillin, clarithromycin, and proton pump inhibitors. Due to the severely increasing global antibiotic resistance, in areas where the resistance to clarithromycin is higher than 20%, the efficacy of triple therapy containing amoxicillin and clarithromycin has been lower than 80% [33]. This treatment is no longer suitable for unconditional empirical use. *H. pylori* biofilms are wrapped with polysaccharides, proteins, extracellular DNA (eDNA), etc., and these external substances protect internal bacteria from external threats. At the same time, the gene expression of the drug efflux pump is significantly increased compared with planktonic cells, which makes it difficult for antibiotics to achieve the desired therapeutic effect. Whether it increases antibiotic resistance or the protective effect of biofilms, there is an urgent need to improve the therapeutic effect of triple therapy.

The MICs of ten *H. pylori* strains (eight clinical strains, SS1 and ATCC43504) for five clinical antibiotics (amoxicillin, clarithromycin, levofloxacin, tetracycline, and metronidazole) showed that metronidazole, levofloxacin, and clarithromycin resistance were higher than amoxicillin and tetracycline resistance. This is consistent with the fact that the resistance rate of *H. pylori* to clarithromycin, metronidazole, and levofloxacin in China is high and has increased over time, while the resistance rate of amoxicillin and tetracycline is low and has been stable over time [34]. In China, bismuth-containing quadruple therapy is recommended as the first-line treatment for *H. pylori*. This therapy is generally effective in China, but studies have suggested that bismuth-containing quadruple therapy containing clarithromycin, metronidazole, and levofloxacin may be inappropriate, and antimicrobial susceptibility testing may be required before treatment. It is recommended that antibiotics with lower resistance rates (amoxicillin and tetracycline) be used in bismuth-containing quadruple therapy [35]. The high resistance rate of clarithromycin may be attributed to the increased consumption of macrolide drugs and the widespread use of metronidazole in the treatment of anaerobic infections (oral, gastrointestinal, etc.) [34]. Although the latest international consensus report strongly recommends the selection of treatment methods based on local drug resistance patterns, testing is rarely performed [4]. It should be noted that the clinical strains used in this experiment lacked patient information, so the resistance cannot be accurately determined with respect to primary or secondary.

In the experiment, ten *H. pylori* strains were treated with amoxicillin and clarithromycin, and the FICs showed that the two antibiotics had partially synergistic, additive, antagonistic, and indifferent relationships among the 10 strains, which may indirectly explain the different therapeutic effects of the same medication for clinical treatment. Every *H. pylori* strain is specific for the same antibiotics, especially for antagonization, and it may be necessary to increase the amount of antibiotics for a better treatment effect. Different FICs may explain the failure of clinical treatment with amoxicillin and clarithromycin, and we recommend testing for drug sensitivity before treatment. This is in line with Yang’s discovery that *Bifidobacterium breve* YH68 and different antibiotic combinations have different interactions with *C. difficile* [24]. In this study, amoxicillin and clarithromycin were used in the treatment of the *H. pylori* biofilm with the FIC concentration, referring to Yang’s use of the antibiotic FIC as the concentration when *Bifidobacterium breve* YH68 was combined with different antibiotics, which was different from the actual ratio of the two antibiotics in clinical triple therapy [22,24,36,37,38]. It is recommended that amoxicillin, clarithromycin, and lansoprazole be administered at doses of 1000 mg, 500 mg, and 30 mg each time by the Fifth Chinese National Consensus Report on the Management of *H. pylori* Infection and that clarithromycin and amoxicillin should be used at a ratio of 2:1. However, it was difficult to accurately determine the appropriate doses of amoxicillin and clarithromycin in this study [22,24,39].

Clarithromycin can bind to the 50S subunit of the ribosome and inhibit protein synthesis, and antibacterial effects are exerted by transforming bacteria from the reproductive phase to the stationary phase [38]. Amoxicillin can inhibit the synthesis of bacterial cell walls during the reproductive stage, and a large amount of water penetrates into the cells, eventually causing bacterial rupture and death [37]. Theoretically, based on pharmacodynamics, when clarithromycin rapidly inhibits the multiplication of *H. pylori*, the bactericidal effect of amoxicillin will be greatly weakened outside this period, and the combination of these two antibiotics will have an antagonistic effect. This explains the antagonization between amoxicillin and clarithromycin.

*H. pylori* can form biofilms at the air–liquid interface in Brucella broth (BB) containing 2–10% (*v*/*v*) fetal bovine serum (FBS). Studies have shown that *H. pylori* can form more biofilms in BB containing 2% FBS [37]. In our research, two *H. pylori* strains had stronger abilities to form biofilms, but the resistance of these two strains to five antibiotics was not significantly different from that of the other eight strains, and the biofilm biomass formed by resistant and sensitive strains was relatively equal. This result was in accordance with the results obtained by Fauzia on 101 clinical isolates [15]. We believe that the MIC cannot be used to predict the biofilm-forming ability of different strains. Planktonic antibiotic-sensitive and antibiotic-resistant strains may have equal abilities to form biofilms.

The MBEC is used as an index to evaluate the sensitivity of biofilms to antibiotics. After measuring the MBEC of 3192 biofilms, it was found that the structure of 3192 biofilms can significantly improve the resistance to amoxicillin and clarithromycin, as the amoxicillin and clarithromycin MBECs were 256-fold and 8-fold higher, respectively, which is consistent with existing reports [36,40]. *H. pylori* biofilms can increase resistance to antibiotics. At present, amoxicillin resistance is not as serious as clarithromycin resistance. We chose clinical strain 3192 as the research object, which has a strong ability to form biofilms and is convenient for research. At the same time, the FIC showed that clinically used amoxicillin and clarithromycin showed antagonism for 3192, but when 3192 forms a biofilm in vivo, treatment requires a higher concentration of amoxicillin on one hand, and on the other hand, if clarithromycin is used in combination with the drug, the weakening of each other’s effectiveness is also the cause of the failure. As 3192 has higher requirements for treatment, it helps to screen superior probiotic strains in vitro. With reference to *Staphylococcus*, we also recommend adding MBEC measurements during *H. pylori* treatment, which can support clinicians in improving treatment options, especially in the case of treatment failure [15].

As a part of the human ecosystem, microbial community imbalance is related to many pathological processes. With the development of sequencing technology, it has been found that there is also a bacterial ecosystem in the stomach [41,42]. There is continuous evidence that *H. pylori* infection can reshape the gastrointestinal flora through host–microbe interaction or microbe–microbial interaction. *H. pylori* an affect the microbial environment in the stomach by interfering with the expression of the proton pump and competing with other microorganisms. In addition, the immune response triggered by *H. pylori* may affect other microbial members [43,44]. Compared with *H. pylori*-negative patients, positive individuals have a unique intestinal community, which indicates that there is an interaction between the gastrointestinal segments. Although there is still a lack of precise basic mechanisms, possible explanations are constantly emerging [45]. Oh et al. found that after standard triple therapy (clarithromycin, amoxicillin, and lansoprazole), the relative abundances of Firmicutes were reduced and Proteobacteria were increased in the human gut microbiota, so *Bacillus* and *Streptococcus* supplements can reduce these changes and imbalances, may limit the growth of clarithromycin and amoxicillin resistant bacteria (*Citrobacter*, *Klebsiella*, *Pseudomonas*, and *Escherichia*) in the intestine by 16S rRNA gene-pyrosequencing, and increase the success rate of *H. pylori* eradication [46,47].

The two core goals of *H. pylori* treatment are to improve the eradication rate and reduce the adverse effects of drugs. Adverse effects of antibiotics (such as diarrhea, nausea, and vomiting) and costly treatments lead to reduced patient compliance. In the *H. pylori* eradication program, it is recommended that probiotics can be used as adjuvants with clinical therapy to reduce the adverse effects of drugs, improve patient compliance, and increase eradication rates. Some current guidelines have recommended the use of probiotics (*Lactobacillus* and *Bifidobacterium*) as adjuvants for eradication. The potential mechanisms include competition for receptors, strengthening the gastric mucosal barrier, inhibiting colonization and adhesion, regulating the immune response of *H. pylori*, and copolymerizing with *H. pylori* to reduce the incidence of adverse reactions and subsequently improve compliance [27,48]. However, it is still necessary to clarify which specific probiotic strains to use, the dosage of probiotics, the timing of administration, and the course of treatment in different geographical areas [19]. This paper is the first to study the effects of the supernatant of *Lactobacillus salivarius* in combination with amoxicillin and clarithromycin on the mature biofilm of *H. pylori* 3192 in vitro. Referring to Yang, we set *Lactobacillus salivarius* LN12 CFS at 1/2 MIC, as at this concentration, CFS has a destructive effect and will not affect the observation of experimental indicators if the concentration is too high [24]. This study found that the killing effect of *Lactobacillus salivarius* LN12 CFS on *H. pylori* biofilms was supplemented by heat-stable short peptides dominated by organic acids, which was consistent with the reported effect of the *Lactobacillus plantarum* LN66 supernatant on *H. pylori* ATCC43504 biofilms. Probiotics can secrete antibacterial substances such as lactic acid, short-chain fatty acids, and bacteriocins [49,50]. After evaluating the biomass, survival rate, protein, and EPS of the biofilm and examination through SEM and CLSM, it was found that *Lactobacillus salivarius* LN12 CFS, amoxicillin, and clarithromycin had a synergistic effect on the mature *H. pylori* biofilm, and the effect of CFS alone was greater than the FIC in terms of the survival rate and EPS content. We believe that *Lactobacillus salivarius* LN12 has better potential application value with amoxicillin and clarithromycin in eradicating *H. pylori* biofilms. At present, *Lactobacillus salivarius* is rare in the treatment of *H. pylori* [51,52].

*CagA* and *vacA gene*, as the two main virulence genes of *H. pylori*, can not only affect the colonization of *H. pylori* in vivo, but also activate inflammatory factors (including IL-8 and TNF-α) to induce the NF-κB signaling pathway related to cytotoxic cell apoptosis [53,54]. Sodium butyrate can inhibit the growth of *H. pylori* and reduce the mRNA expression of *H. pylori cagA* and *vacA* in vitro [55]. This study found that *Lactobacillus salivarius* LN12 CFS, amoxicillin, and clarithromycin alone or in combination all downregulate *vacA* gene expression, and the downregulation effect of the combined use was greater than the effect of their single actions. However, the *cagA* gene was downregulated in the antibiotics treatment group, but was upregulated in the LN12 CFS supernatant treatment group, while there was no significant difference between the CF treatment group and the control group. Usually, bacteria downregulate growth and proliferation genes to survive in an unfavorable environment, and the transcription of genes related to survival and virulence is upregulated. Studies have shown that the synergy of resveratrol and ethanol can upregulate the gene transcription of *cagA*, urease, and other virulence factors of *H. pylori* 26695 in vitro, and upregulation of genes involved in antioxidation and maintenance of iron homeostasis will facilitate the survival of resveratrol under *H. pylori* oxidative stress [56,57]. Since the gene expression process of pyloric biofilms is not fully explained at present, only two main virulence factors were selected in this study. The effects of *Lactobacillus salivarius* LN12 CFS in combination with amoxicillin and clarithromycin on the other virulence factors of biofilms will be revealed by subsequent transcriptome analysis.

There are different modeling methods for culturing *H. pylori* biofilms such as static culture, liquid culture, and microfluidic culture. For the microporous culture method, *H. pylori* is inoculated in a cell culture plate containing medium, where biofilms form at the air–liquid interface, and the amount of biofilm is relatively small, which is affected by the specificity of the bacteria and the medium conditions [58,59]. On the other hand, a colony biofilm is formed when a nitrocellulose membrane or other carriers are placed on the plate medium, and the bacteria absorb nutrients through the membrane to form a biofilm structure on the surface of the membrane [60]. Ji used the colony biofilm modeling method to study the effect of *Lactobacillus plantarum* LN66 CFS combined with CLR or levofloxacin (LVX) on *H. pylori* ATCC43504 mature biofilms and found that CFS, CLR, and LVX had killing effects on *H. pylori* cells and can also significantly reduce the expression levels of the *H. pylori cagA* and *vacA* genes separately. At the same time, CFS with every antibiotic had a synergistic effect on the inhibition of virulence gene expression [29]. Our study also found that the combination of LN12 CFS and AMX and CLR can synergistically downregulate the *vacA* gene, but the change in the *cagA* gene was different from that in Ji’s study, possibly because of the different modeling methods.

## 5. Conclusions

In the present study, we first explored the destructive effect of *Lactobacillus salivarius* LN12 CFS in combination with amoxicillin and clarithromycin on the biofilm of *H. pylori* 3192 in vitro. Clarithromycin and amoxicillin showed antagonistic effects on planktonic 3192 in vitro, and 3192 had a strong biofilm-forming ability. The destructive effect of LN12 CFS on the biofilm was dominated by heat-stable short peptides supplemented with organic acids. The combination had more significant effects on biomass, survival rate, and structure damage of the biofilm than that of the single use. At the same time, it also had a significant downregulation effect on the virulence gene *vacA*. The clarifying mechanism of the combination on the biofilm and the dosage of probiotics when used in combination requires follow-up research.

## Figures and Tables

**Figure 1 microorganisms-09-01611-f001:**
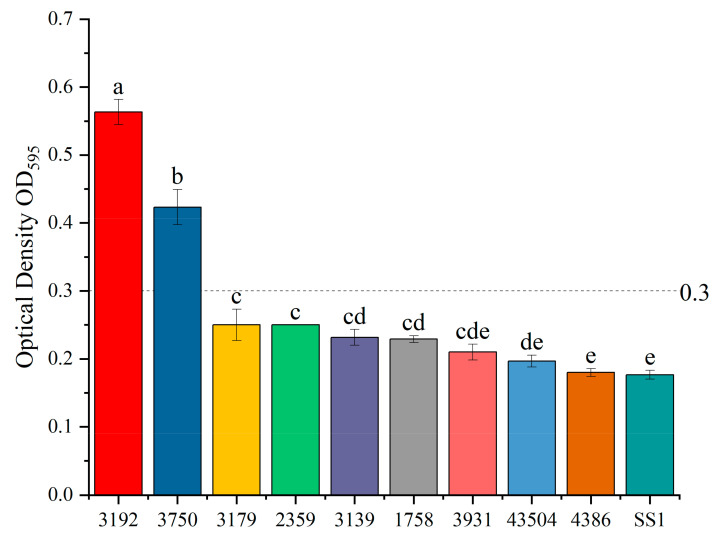
Distribution of biofilm formation in ten *H. pylori* strains. The biofilm-forming abilities of ten *H. pylori* strains were compared by crystal violet staining, and biofilm biomass was measured according to OD_595nm_. OD_595nm_ 0.3 was regarded as the dividing line. All of the results are expressed as the means ± standard deviation from three independent experiments. The groups marked with different superscript letters indicate statistically significant differences (*p* < 0.05).

**Figure 2 microorganisms-09-01611-f002:**
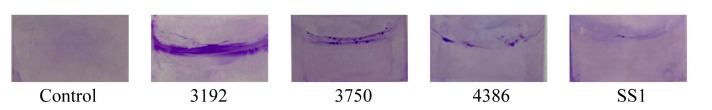
Biofilm formation by four *H. pylori* strains on glass coverslips after four days in BB2. *H. pylori* strains: 3192, 3750, 4386, SS1. Biofilm was formed at the gas–liquid interface on 20 mm × 20 mm borosilicate coverslips and stained with 1% crystal violet solution.

**Figure 3 microorganisms-09-01611-f003:**
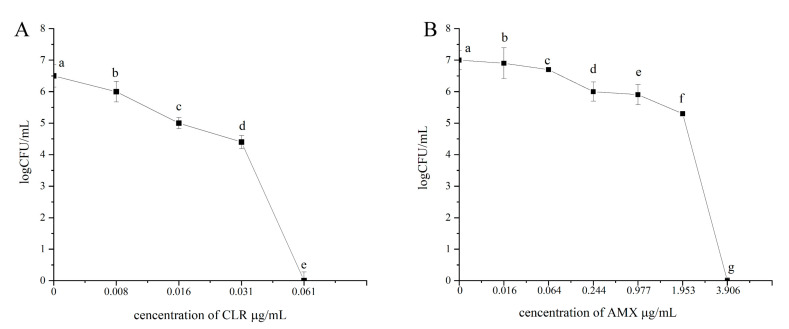
The effects of CLR (**A**) and AMX (**B**) on the cell viability of *H. pylori* 3192 biofilms. Biofilms of 3192 were exposed to different concentration of CLR and AMX. After exposure to the antibiotics, viable cells from biofilms were measured using CFU counting. All of the results are expressed as the means ± standard deviation from three independent experiments. The groups marked with different superscript letters indicate statistically significant differences (*p* < 0.05). AMX: Amoxicillin, CLR: Clarithromycin.

**Figure 4 microorganisms-09-01611-f004:**
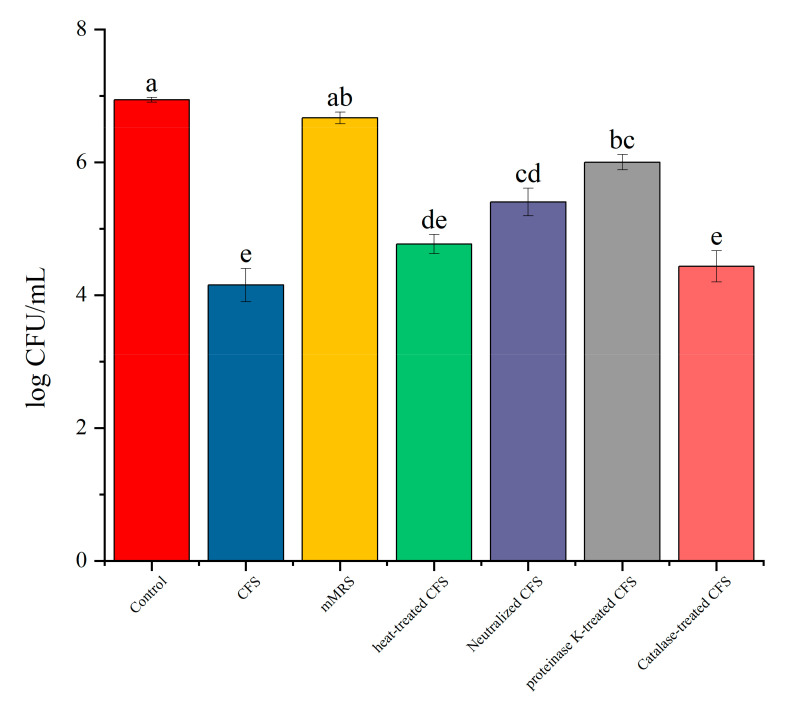
Effects of LN12 CFS with different treatments on the *H. pylori* 3192 biofilm. Heat-treated CFS: LN12 CFS 115 °C heated for 20 min; neutralized CFS: LN12 CFS adjusted the pH to 6.5 with 1 M NaOH; proteinase K-treated CFS: LN12 CFS treated with 1 mg/mL proteinase K (final concentration) for 1 h at 37 °C and inactivated by heating at 95 °C for 1 min; catalase-treated CFS: LN12 CFS treated with 0.5 mg/mL catalase (final concentration) for 1 h at 37 °C and inactivated by heating at 95 °C for 30 min; mMRS: modified de Man Rogosa Sharpe broth; BB2 were the controls. The experimental data are represented by the mean ± standard deviation (*n* = 3), and the groups marked with different superscript letters indicate statistically significant differences (*p* < 0.05).

**Figure 5 microorganisms-09-01611-f005:**
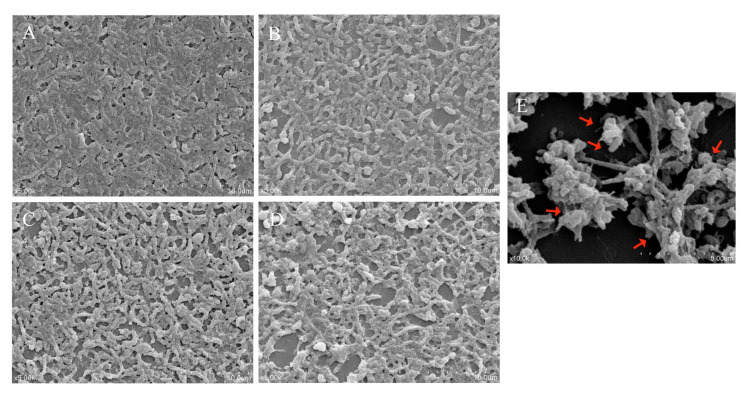
SEM of *H. pylori* 3192 biofilm on the coverslip. (**A**,**E**): Control, (**B**): 1×FIC of AMX and CLR treatment, (**C**): 1/2 × MIC of LN12 CFS treatment, (**D**): 1/2 × MIC of LN12 CFS in combination with 1 × FIC of AMX and CLR treatment. The magnification is 5000× for **A**–**D**, the scale bar is 10.00 μm; 10,000× for **E**, and the scale bar is 5.00 μm. AMX: Amoxicillin, CLR: Clarithromycin.

**Figure 6 microorganisms-09-01611-f006:**
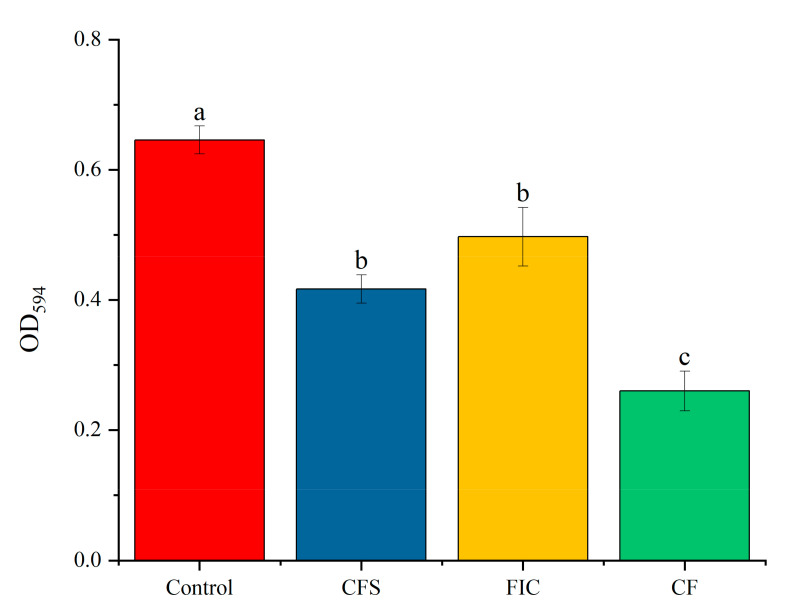
Biofilm biomass of thee *H. pylori* 3192 biofilm under different treatments. CFS: 1/2 × MIC of LN12 CFS treatment, FIC: 1 × FIC of AMX and CLR treatment, CF: 1/2 × MIC of LN12 CFS in combination with 1 × FIC of AMX and CLR treatment. The experimental data are represented by the mean ± standard deviation (*n* = 3), and the groups marked with different superscript letters indicate statistically significant differences (*p* < 0.05). AMX: Amoxicillin, CLR: Clarithromycin.

**Figure 7 microorganisms-09-01611-f007:**
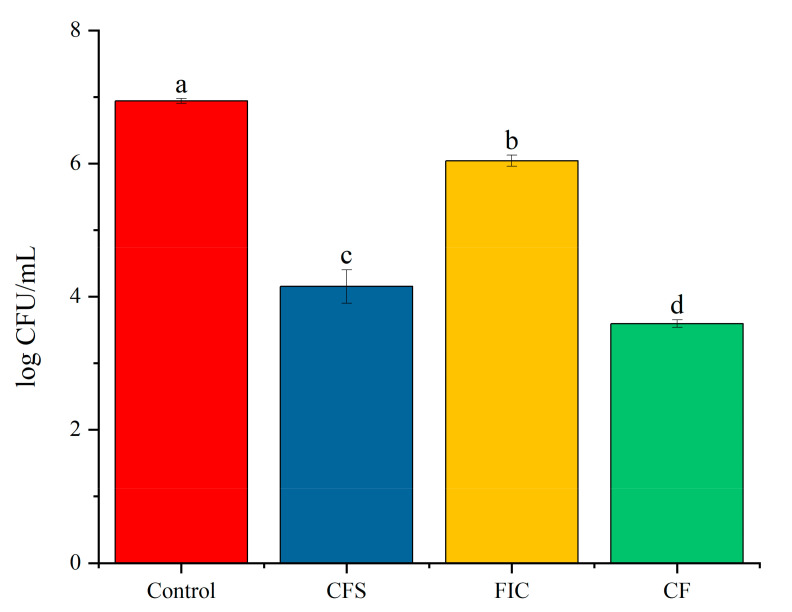
Survival rate of the *H. pylori* 3192 biofilm under different treatments. CFS: 1/2 × MIC of LN12 CFS treatment, FIC: 1 × FIC of AMX and CLR treatment, CF: 1/2 × MIC of LN12 CFS in combination with 1 × FIC of AMX and CLR treatment. The experimental data are represented by the mean ± standard deviation (*n* = 3), and the groups marked with different superscript letters indicate statistically significant differences (*p* < 0.05). AMX: Amoxicillin, CLR: Clarithromycin.

**Figure 8 microorganisms-09-01611-f008:**
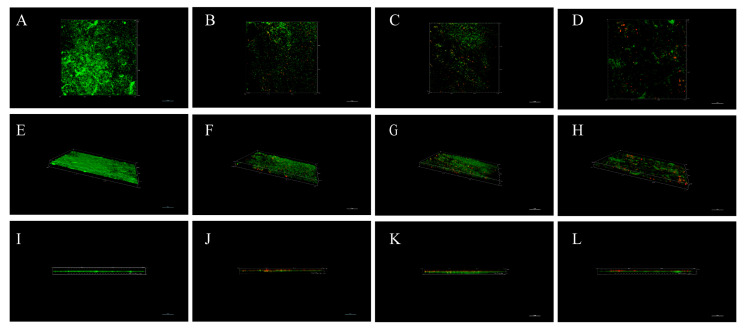
CLSM of the *H. pylori* 3192 biofilm under different treatments. (**A**–**D**) was the main view (observed horizontally along the x–y axis, **E**–**H**) was the view at a certain angle, (**I**–**L**) was the side view (observed horizontally along the x–z axis). Control: (**A**,**E**,**I**); 1/2 × MIC of LN12 CFS treatment: (**B**,**F**,**J**); 1 × FIC of AMX and CLR treatment: (**C**,**G**,**K**); 1/2 × MIC of LN12 CFS in combination with 1 × FIC of AMX and CLR treatment: (**D**,**H**,**L**). The scale bar is 10 μm. AMX: Amoxicillin, CLR: Clarithromycin.

**Figure 9 microorganisms-09-01611-f009:**

CLSM of the *H. pylori* 3192 biofilm EPS under different treatments. (**A**) Control, (**B**) 1/2 × MIC of LN12 CFS treatment, (**C**) 1 × FIC of AMX and CLR treatment, (**D**) 1/2 × MIC of LN12 CFS in combination with 1 × FIC of AMX and CLR treatment. The scale bar is 20 μm. AMX: Amoxicillin, CLR: Clarithromycin.

**Figure 10 microorganisms-09-01611-f010:**
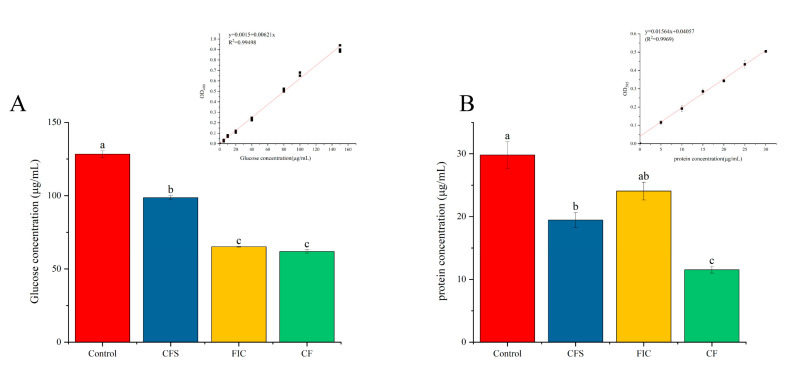
Different treatments on polysaccharide (**A**) and protein (**B**) content in the *H. pylori* 3192 mature biofilm. CFS: 1/2 × MIC of LN12 CFS treatment, FIC: 1 × FIC of AMX and CLR treatment, CF: 1/2 × MIC of LN12 CFS in combination with 1 × FIC of AMX and CLR treatment. The experimental data are represented by the mean ± standard deviation (*n* = 3), and the groups marked with different superscript letters indicate statistically significant differences (*p* < 0.05). AMX: Amoxicillin, CLR: Clarithromycin.

**Figure 11 microorganisms-09-01611-f011:**
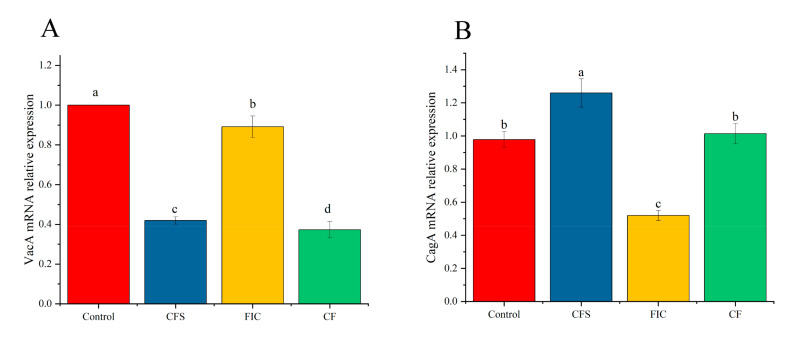
Different treatments on the expression levels of *vacA* (**A**) and *cagA* (**B**) genes in the *H. pylori* 3192 biofilm. CFS: 1/2 × MIC of LN12 CFS treatment, FIC: 1 × FIC of AMX and CLR treatment, CF: 1/2 × MIC of LN12 CFS in combination with 1 × FIC of AMX and CLR treatment. The experimental data are represented by the mean ± standard deviation (*n* = 3), and the groups marked with different superscript letters indicate statistically significant differences (*p* < 0.05). AMX: Amoxicillin, CLR: Clarithromycin.

**Table 1 microorganisms-09-01611-t001:** Real-time PCR primers.

Genes	Sequences (5′–3′)
*cagA*-F	TGGCTAAAGCAACGGGTGAT
*cagA*-R	CCGACTAGGGTTCCGTTCAC
*vacA*-F	CACTAACGCTGATGGCACGA
*vacA*-R	GGACAGATTGACACCGCCTT
16s rRNA-F	GGCGACCTGCTGGAACATTACTG
16s rRNA-R	CATCGTTTAGGGCGTGGACTACC

**Table 2 microorganisms-09-01611-t002:** Minimal inhibitory concentration of antibiotics for *H. pylori* isolates.

*H. pylori* Isolates	Minimal Inhibitory Concentration (mg/L)				
	Amoxicillin	Clarithromycin	Levofloxacin	Metronidazole	Tetracycline
SS1	0.094(S)	0.032(S)	0.094(S)	0.032(S)	0.032(S)
ATCC43504	0.016(S)	0.032(S)	0.19(S)	256(R)	0.19(S)
3192	0.016(S)	0.016(S)	8(R)	256(R)	0.125(S)
3750	0.094(S)	24(R)	32(R)	256(R)	0.38(S)
3139	0.016(S)	0.016(S)	0.25(S)	12(R)	0.032(S)
4386	0.25(R)	16(R)	0.5(S)	256(R)	0.75(S)
2359	0.016(S)	0.016(S)	0.094(S)	1(S)	0.023(S)
1758	0.023(S)	2(R)	32(R)	24(R)	0.023(S)
3931	0.016(S)	12(R)	32(R)	256(R)	0.25(S)
3179	0.016(S)	0.023(S)	32(R)	256(R)	0.25(S)

S: susceptible; R: resistant.

**Table 3 microorganisms-09-01611-t003:** Fractional inhibitory concentrations (FICs) of amoxicillin and clarithromycin on *H. pylori*.

Strain	FIC s (MIC_CLR_, MIC_AMX_)	Effect
SS1	1.5 (0.5, 1)	Indifferent effect
ATCC43504	0.75 (0.25, 0.5)	Partially synergistic
3192	2.125 (2, 0.125)	Antagonistic
3750	1.06 (0.06, 1)	Indifferent effect
3139	1.0 (0.5, 0.5)	Indifferent effect
4386	1.06 (0.06, 1)	Indifferent effect
2359	0.625 (0.125, 0.5)	Partially synergistic
1758	1.0 (0.5, 0.5)	Indifferent effect
3931	2.25 (0.25, 2)	Antagonistic
3179	0.625 (0.125, 0.5)	Partially synergistic

AMX: Amoxicillin, CLR: Clarithromycin.

**Table 4 microorganisms-09-01611-t004:** Anti-*H. pylori* activity of CFSs collected from seven probiotics.

Probiotic Strains	Diameter of Inhibition Zone/mm									
	SS1	43504	3750	4386	3192	3931	3179	3139	2359	1758
LN66	6.8 ± 0.3 a	8.1 ± 0.4 a	6.6 ± 0.6 a	6.7 ± 0.3 a	5.2 ± 0.3 a	5.8 ± 0.3 a	5.7 ± 0.7 a	5.5 ± 0.5 ab	5.8 ± 0.2 b	7.5 ± 0.3 a
LN12	5.6 ± 0.6 b	8.1 ± 0.3 a	5.1 ± 0.4 b	5.8 ± 0.2 b	4.7 ± 0.3 a	5.5 ± 0.1 ab	5.3 ± 0.2 ab	5.8 ± 0.3 a	6.7 ± 0.4 a	6.4 ± 0.2 b
LN19	5.3 ± 0.5 b	8.0 ± 0.3 a	4.8 ± 0.2 bc	5.2 ± 0.3 b	4.8 ± 0.3 a	4. ± 0.2 bcd	5.7 ± 0.4 a	4.8 ± 0.2 bc	3.1 ± 0.2 d	4.9 ± 0.1 c
INO-10	5.5 ± 0.2 b	6.7 ± 0.2 b	3.9 ± 0.4 cd	3.60 ± 0.2 c	3.5 ± 0.2 b	5. ± 0.8 abc	4.2 ± 0.2 bc	3.8 ± 0.2 de	4.0 ± 0.3 c	0 e
INO-17	3.6 ± 0.2 c	5.5 ± 0.3 c	3.9 ± 0.4 cd	5.2 ± 0.2 b	3.50 ± 0.7 b	4.1 ± 0.3 cd	0 d	3.2 ± 0.3 e	3.1 ± 0.2 d	3.1 ± 0.1 d
INO-31	3.8 ± 0.5 c	5.3 ± 0.4 c	3.1 ± 0.3 d	3.8 ± 0.6 c	3.77 ± 0.1 b	3.3 ± 0.3 d	0 d	3.3 ± 0.2 e	3.6 ± 0.3 cd	4.6 ± 0.3 c
INO-11	4.6 ± 0.2 bc	6.8 ± 0.4 b	3.8 ± 0.2 d	0 d	3.3 ± 0.2 b	3.5 ± 0.5 d	3.6 ± 0.6 c	4.4 ± 0.2 cd	3.5 ± 0.3 cd	3.2 ± 0.2 d
mMRS	0	0	0	0	0	0	0	0	0	0

Probiotic strains: LN66 (*Lactobacillus plantarum*), LN12 (*Lactobacillus salivarius*), LN19 (*Lactobacillus helveticus*), INO-10, INO-17, INO-31, INO-11 (*Bifidobacterium longum*). *H. pylori* strains: SS1, 43504, 3750, 4386, 3192, 3931, 3179, 3139, 2359, 1758. All of the results are expressed as the means ± standard deviation from three independent experiments. The groups marked with different letters indicate statistically significant differences (*p* < 0.05).

## Data Availability

Data is contained within the article and Appendix A.

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
