# Peer review of "Effects of Lactobacillus salivarius LN12 in Combination with Amoxicillin and Clarithromycin on Helicobacter pylori Biofilm In Vitro"

_microorganisms, 2021, doi:10.3390/microorganisms9081611_

Round 1
Reviewer 1 Report
In this manuscript, Jin and Yang studied the effects of Lactobacillus salivarius LN12 combined with antibiotics on Heliobacter pylori biofilm. The Authors used both traditional culture-dependent methods combined with biofilm visualization and qPCR genetic study of virulence gene expression. In my opinion, the paper is well structured and written. The experiment is well-designed and it is very interesting. However, I have few suggestions which can improve the paper:
- Introduction section: I suggest adding short paragraph about the mechanisms and genes that participated in the virulence mechanisms of H. pylori
- Line 212 – you studied the RNA so it should be: „on the expression of virulence gene…”
- Section 2.12 – Did the Authors tested the quantity and quality of RNA after the extraction process?
- Figure 1 – please add a description to the x-axis
- Figure 2 – pictures are very small and unreadable
- Figure 4 – X-axis labels are too small and unreadable
Reviewer 2 Report
This manuscript by Fang Jin and Hong Yang describes very interetsing data on the effect of Lactobacillus strains onto Hp biofilm formation in association with two major antibiotics.
This manuscript deserves publication after minor revision.
Global : italicize "in vivo", "in vitro", gene/bacterial names.
Introduction/Discussion. As the authors suggest the possible use of probiotic strains on the infection by Hp, they have to discuss the impact of Hp/their intervention on GIT microbiome (see Pichon and Burucoa Healthcare for example).
Table 2 : Indicate classification S/I/R in the table near to the MIC value, to ease the reading.
Table 3. How do the authors explain the very different results between FIC of Cla and Amox onto these strains?
Typo have to be corrected (see for example table 4). Moreover, the police has to be uniformized sometimes. Finally, in table and in figure, prefer indicate significant results by signs (in table) or p-values (in figure) more than in "different superscript". Moreover, comparison have to be performed two-by-two for all type of treatment for table 5-6-7
Author Response
Response to Reviewer 2 Comments
Point 1: Global : italicize "in vivo", "in vitro", gene/bacterial names.
Response 1: Thank you for your suggestion, all fonts were revised in this revised manuscript.
Point 2: As the authors suggest the possible use of probiotic strains on the infection by Hp, they have to discuss the impact of Hp/their intervention on GIT microbiome (see Pichon and Burucoa Healthcare for example).
Response 2: Thank you for your suggestion and we are agreed with you. A paragraph about the impact of Hp/Hp-probiotics intervention on GIT microbiome was added to the discussion(line554-568).
Point 3: Indicate classification S/I/R in the table near to the MIC value, to ease the reading.
Response 3: Thank you for your suggestion, we determined the MIC value of five commonly used antibiotics in clinical treatment (amoxicillin, clarithromycin, levofloxacin, tetracycline, and metronidazole) for 10 H. pylori strains. If this value was lower than the prescribed resistance cut-off value according to the recommendations of the European Antimicrobial Susceptibility Testing Committee (EUCAST), the bacteria was considered to be sensitive to
the antibiotic(S), if it is higher than the cut-off value, it is considered resistant to this antibiotic(R),we did not use the definition of Intermediary(I),we indicated classification S/R near to the MIC value in the Table 2. Point 4: Table 3. How do the authors explain the very different results between FIC of Cla and Amx on these strains?
Response 4: In the experiment, 10 H. pylori strains were treated with amoxicillin and clarithromycin, and the FICs showed that the two antibiotics had partially synergistic, additive, antagonistic and indifferent relationships among the 10 strains. This result was very interesting, two explanations were given in the discussion section:1. H. pylori was strain-specific, and each
strain had its own uniqueness; 2.Clarithromycin can bind to the 50S subunit of the ribosome and inhibit protein synthesis, and antibacterial effects are exerted by transforming bacteria from the reproductive phase to the stationary phase. Amoxicillin can inhibit the synthesis of bacterial cell walls during the reproductive stage, and a large amount of water penetrates into the cells,
eventually causing bacterial rupture and death. Theoretically, based on pharmacodynamics, when CLR rapidly inhibits the multiplication of H. pylori, the bactericidal effect of AMX will be greatly weakened outside this period, and the combination of these two antibiotics will have an antagonistic effect. This explains the antagonization between amoxicillin and clarithromycin(line 520-528).
Point 5: Typo have to be corrected (see for example table 4).
Response 5: Thank you for your suggestion, typo in table 4 have been corrected .
Point 6: Moreover, the police has to be uniformized sometimes. Finally, in table and in figure, prefer indicate significant results by signs (in table) or p-values (in figure) more than in "different superscript". Moreover, comparison have to be performed two-by-two for all type of
treatment for table 5-6-7
Response 6: Thank you for your suggestion and we are agreed with you. We think that different superscript letters can make the significant differences clearer especially in Figure 4 and Table 4, we use this uniform police throughout the paper. For all types of treatments in Figure 5-6-7, comparisons was performed two-by-two
Reviewer 3 Report
Very interesting work with promising results. We are all well aware of the challenge of treating H. pylori infections so research into a combination of different antibiotics or other substances, including probiotics, is welcome and commendable.
My comments follow:
line 232 - antibiotics act on bacteria, not the other way around so fix the sentence.
table 2. reduce the font
Fig. 1 lacks a description of the display of results.
it is not clear to me what Fig. 3 shows because you wrote the reaction kinetics or velocity curve. However, these results do not show this. showing the effect of different conc. of antibiotics on the number of bacteria. The result should be presented differently.
the term viability is used for cells that are viable, not necessarily culturable while you were determining cfu, meaning culturable. Or did I misunderstand?
Table 4. Make sure that the names of the bacteria are written in italic.
Fig 4. describe the display of results.
Author Response
Response to Reviewer 3 Comments
Point 1: line 232 - antibiotics act on bacteria, not the other way around so fix the sentence.
Response 1: Thank you for your suggestion and we are agreed with you. The sentence was corrected as followed: the MICs of amoxicillin, clarithromycin, levofloxacin, tetracycline, and metronidazole for 10 H. pylori strains were shown in Table 2(line 243),the line number was changed because two paragraphs were added to the introduction and discussion., separately.
Point 2: table 2. reduce the font
Response 2: Thank you for your suggestion and we are agreed with you. The format of the table was adjusted according to the requirements of the journal, and the font size was reduced in Table 2.
Point 3: Fig. 1 lacks a description of the display of results.
Response 3: Thank you for your suggestion and we are agreed with you. We added a title to x-axis, and a description of the display of results to Figure. 1.
Point 4: it is not clear to me what Fig. 3 shows because you wrote the reaction kinetics or velocity curve. However, these results do not show this. showing the effect of different conc. of antibiotics on the number of bacteria. The result should be presented differently. the term viability is used for cells that are viable, not necessarily culturable while you were determining cfu, meaning culturable. Or did I misunderstand?
Response 4: Thank you for your suggestion, we are agreed with you. Colony forming unit (CFU) refers to the colony formed by the growth and reproduction of a single bacterial cell or multiple bacterial cells aggregated on a solid medium during the culture and counting of live bacteria. It is called a colony forming unit, which expresses the number of live bacteria. We also think that it was inappropriate to use the description of reaction kinetics or velocity curve.
Referencing to related research ,we revised the title as: The effects of CLR(A) and AMX(B) on cell viability of H. pylori 3192 biofilms[1]. The content of the figure remains unchanged. Figure 3 showed the changes of cell viability in 3192 H. pylori biofilms under different gradient concentrations of CLR and AMX for 24 hours, We hope that the minimum biofilm removal concentration(MBEC) can be found from the relationship between the antibiotic concentration
and the CFU in the biofilm. MBEC can reflect the change in antibiotic resistance in biofilm compared to planktonic form. We believe all H. pylori cells in biofilm were viable and culturable on columbia blood agar plates containing 5% sterile defibrinated sheep blood for 48-72 h referring to existing reports[1].
Reference
1. Yonezawa, H.; Osaki, T.; Hojo, F.; Kamiya, S. Effect of Helicobacter pylori biofilm formation on susceptibility to amoxicillin, metronidazole and clarithromycin. Microb. Pathog. 2019, 132, 100-108, doi:10.1016/j.micpath.2019.04.030.
Point 5: Make sure that the names of the bacteria are written in italic
Response 5: Thank you for your suggestion, all fonts were revised in this revised manuscript.
Point 6: Fig 4. describe the display of results.
Response 6: Thank you for your suggestion and we are agreed with you. A description of the display of results was attached under Figure 4(line 336-344)